# Surface Treatments’ Influence on the Interfacial Bonding between Glass Fibre Reinforced Elium^®^ Composite and Polybutylene Terephthalate

**DOI:** 10.3390/ma17133276

**Published:** 2024-07-03

**Authors:** Ashish Matta, Venkat Reddy Yadavalli, Lukas Manas, Marketa Kadleckova, Vladimir Pavlinek, Tomas Sedlacek

**Affiliations:** 1Department of Polymer Engineering, Faculty of Technology, Tomas Bata University in Zlin, Vavreckova 5669, 760 01 Zlin, Czech Republic; matta@utb.cz; 2Department of Applied Logistics and Polymer Sciences, Hochschule Kaiserslautern, University of Applied Sciences, 67659 Pirmasens, Germany; venkatyadavalli.5@gmail.com; 3Centre of Polymer Systems, Tomas Bata University in Zlin, Trida Tomase Bati 5678, 760 01 Zlin, Czech Republic; m1_kadleckova@utb.cz (M.K.); sedlacek@utb.cz (T.S.); 4Department of Production Engineering, Faculty of Technology, Tomas Bata University in Zlin, Vavreckova 5669, 760 01 Zlin, Czech Republic; 5Department of Physics and Materials Engineering, Faculty of Technology, Tomas Bata University in Zlin, Vavreckova 5669, 760 01 Zlin, Czech Republic; 65M s.r.o., Na Zahonech 1177, 686 04 Kunovice, Czech Republic; vladimir.pavlinek@5m.cz

**Keywords:** resin transfer moulding, Elium^®^ composite, glass fibre, polymer insert surface treatment, polybutylene terephthalate, insert moulding

## Abstract

This study examines the process of using injection moulding to join two different materials to manufacture bi-component moulded products with improved performance characteristics. The two-component process, which combines the advantages of two different technologies—the high efficiency of the injection moulding process and the excellent mechanical properties of long glass fibre composites produced by resin transfer moulding (RTM) technology—offers a particular advantage and improved applicability of the prepared lightweight products in both the automotive and aerospace sectors. The composite studied here consists of Elium^®^ thermoplastic resin (30%) reinforced with unwoven glass fibre fabric (70%) using the RTM process. The Elium^®^ composite sample is consequently used as an insert overmoulded with polybutylene terephthalate (PBT) homopolymer reinforced with 20% *w*/*w* of short glass fibre through injection moulding. The influence of different mould temperatures and surface treatments on the adhesion between the materials used is investigated by evaluating the mechanical performance using tensile shear strength tests. It was found that while an increase in mould temperature from 40 °C to 120 °C resulted in a doubling of the initial average bond strength between untreated Elium^®^ RTM inserts and overmoulded PBT parts (0.9 MPa), sandblasting the inserts ensured a further tripling of the bond strength of the composites to a value of 5.4 MPa.

## 1. Introduction

The issues of energy consumption and ecological impact are becoming increasingly prominent in the fields of automation and transportation. The weight of automobiles has a significant impact on these factors, making it imperative to utilise materials and production techniques that minimise it. Composite materials are particularly effective in achieving lightweight and efficient parts that are also stronger, more chemically resistant, and corrosion resistant. Concurrently, it provides manufacturers with the potential to enhance the performance of their products and reduce the environmental impact of production and individual components and assemblies [1].

Thermosetting composites (TSCs) are typically made by impregnating continuous fibres, such as glass or carbon, or their fabrics with a low-viscosity thermosetting resin, which is then cured by the application of heat or ultraviolet (UV) light. This initiates a chemical reaction that irreversibly hardens the material. These composites exhibit high strength, stiffness, excellent thermal stability, and heat resistance due to the cross-linked polymer structure. Although the general disadvantage of higher brittleness of TSC due to the high density of cross-linking can be improved by the implementation of selected thermoplastic resins [2] such as polyamide (PA), polypropylene (PP), or polyether ether ketone PEEK, lower manufacturing productivity associated with technologies used, such as resin transfer moulding (RTM) or pultrusion is a limitation for cost saving of final products [3] compared to thermoplastic composites (TPCs), typically made from a thermoplastic resin reinforced with short fibres or by impregnating continuous fibres/fabrics with a high-viscosity thermoplastic melt. Nevertheless, the utilisation of TPC presents the potential for supplementary processing avenues, including the preparation of preforms used in partial component assemblies. The manufacturing of TPC by means of, for instance, film stacking moulding, powder impregnation, or hybrid woven fabric processing results in lower mechanical performance due to the use of short fibres or a higher amount of thermoplastic resin employed for fabric impregnation. However, this is counterbalanced by a higher toughness and impact resistance [4]. In addition, not only can TPC be produced more efficiently, but they can also be remelted and reshaped several times, which is another advantage [5]. Furthermore, even TSCs have lower material costs than often-used high-performance thermoplastics, such as PEEK, polysulfone (PSU), and polyetherimide (PEI). However, they have higher processing costs due to longer cycle times [6]. Therefore, each approach has its own limitations depending on the specific performance requirements and product application. To further develop the potential of composites, it is essential to undertake a systematic road-mapping activity [7].

Recent advances in automotive applications have focused on thermoplastics and their composites. Among these, organo-sheet composites (OSCs) have attracted much attention [8]. The TPC, made by overmoulding OSC with the thermoplastic matrix design details, is advantageously produced by heating and forming impregnated technical textiles directly in an injection mould. As the organic sheet matrix and the overmoulded TP are compatible, there are few adhesion problems at the interface. Although thermoplastic composites offer a number of benefits, they have limitations, such as high prices and inadequate temperature resistance [8,9]. Nevertheless, these types of composites can generally be used for aerospace structural components, interior panels, automotive body panels, building panels, and bridge components or advantageously employed as aerospace brackets, clips, automotive bumpers, and wind turbine blades. On the other hand, thermosets are preferred over thermoplastics in certain applications due to their superior structural properties. Unlike thermoplastics, thermosets exhibit a lower loss of modulus and strength with increasing temperature and have improved creep resistance. However, incorporating assembly and bonding features into designs has proven challenging [10].

Recent improvements in fast-curing TSCs or the invention of reactive TPCs have made their use more practical. Currently, research is focused on creating combinations of thermoset and thermoplastic materials to overcome joining and design complications. Many joining techniques have been investigated, including laser-based hot melt bonding [11], resistance welding, and ultrasonic welding [12]. Welding options for technical fabric composites are limited, and manufacturing complex geometries is a processing nightmare. Therefore, other common methods for joining TPC, such as adhesive bonding, mechanical fastening, simultaneous curing of two composite parts with or without adhesive, hybrid joints (combining adhesive and mechanical bonding), and injection overmoulding are being extensively investigated to meet growing industrial challenges. In particular, the bonding of incompatible surfaces is often studied with a view to increasing the mechanical performance of the joint using various surface treatments. These include mechanical roughness modifications or external high-energy operation via the plasma approach using various gases, such as nitrogen, argon, fluorine, carbon dioxide, and oxygen [13]. In the context of individual applications, the most appropriate surface treatment methods are selected according to the specific industrial process, assembly technology, and combination of materials employed.

The recently introduced Elium^®^ matrix can be processed using a range of methods, including injection moulding, extrusion, and compression moulding, in a manner similar to other thermoplastics. This adaptability enables production flexibility and the fabrication of complicated shapes, providing to a broad spectrum of uses [14]. What distinguishes Elium^®^ is its reactive thermoplastic nature. This implies that to obtain better mechanical qualities, it is cross-linked during processing. The cross-linking mechanism effectively connects the gap between conventional thermoplastics and thermosetting resins, imparting the material with increased strength and rigidity. One significant advantage of Elium^®^ is that it can acquire mechanical characteristics similar to those of thermosets [15]. Moreover, it is frequently combined with reinforcing fibres, such as carbon or glass fibres, for use in composite products. This combination significantly enhances the mechanical qualities of the resin, making it suitable for a wide range of applications, including construction materials, aircraft components, and vehicle parts. Developed by Arkema, Elium^®^ features a unique resin system that allows for both infusion and injection processes, rendering it well-suited for various manufacturing techniques, including resin transfer moulding (RTM). The resin, renowned for its low viscosity and rapid cure characteristics, facilitates efficient impregnation of reinforcing fibres, resulting in high-performance composite parts with outstanding mechanical properties. The application of Elium^®^ in the field of composites broadens its market reach to include sectors where toughness and lifespan are significant [16,17]. Elium^®^ composites are often employed in the fabrication of two real-time components: composite sandwich structures and pultruded sections. The utilisation of Elium^®^ resin in the construction of sandwich structures has demonstrated enhanced flexural and flatwise strength when compared to traditional resins, rendering them suitable for high-performance applications, including wind turbine blades. Elium^®^ resin-pultruded parts exhibit qualities equivalent to ordinary thermoset resins but additionally possess the capacity to be post-formed or shaped while maintaining mechanical performance [18]. Furthermore, thermoplastic tubular composites manufactured from Elium^®^ composites have been demonstrated to exhibit greater resilience to impact, superior flexural qualities, and a reduction in vibrations compared to carbon/epoxy [19]. These findings indicate that Elium^®^ composites have the potential to be utilised in a diverse range of industries, including sports, wind turbines, automobiles, and beyond [20].

Short glass fibre-reinforced polybutylene terephthalate (PBT) is a type of composite material that combines the benefits of glass fibre reinforcement and thermoplastics. When thermoplastics are combined with glass fibres, such composites exhibit improved heat resistance, dimensional stability, and mechanical strength. This combination of PBT and glass fibre retains the favourable properties of PBT, such as high electrical insulation and chemical resistance, in addition to its reduced warpage properties. Fundamentally, the 20% short glass fibre-reinforced PBT blend provides an ideal balance of strength, dimensional stability, and structural integrity, reducing the likelihood of warpage during moulding, making it a flexible option suitable for many different engineering applications.

The present study focuses on the combination of Elium^®^ composites prepared via RTM technology with PBT overmoulded part to produce the bi-component samples. The reason for choosing Elium^®^ and PBT is twofold. Firstly, there is limited information on the combination of these two promising materials, which could potentially bring novelty and offer potential options for bi-component components and products used in the automotive and aerospace industries. Secondly, the combination of long glass fibre-reinforced Elium^®^, and short glass fibre-reinforced PBT has the potential to offer improved properties and mechanical performance that could be exploited in the above-mentioned industries. A variety of surface treatments, including solvent-induced swelling, plasma jetting, and sandblasting, have been used to gain a deeper insight into the effect of the chosen surface treatment process. In this way, the work presented focuses on the in-line assembly of long glass fibre composites and short glass fibre-reinforced engineering materials via mechanical interlocking (see Figure 1) due to its convenience and ability to create strong bonds [21], thus offering competent mechanical performance for the finished bi-component products and structures, potentially usable in various means of transport interior wall assembly or wind blade mounting.

## 2. Materials and Methods

This section describes the material selection for the RTM and overmoulding processes, the preparation of the Elium^®^ sheets reinforced with fibreglass fabrics, including the surface treatment of the prepared composite inserts together with a description of their characterisation, and the fabrication of PBT overmoulded bi-component test specimens, together with a description of their mechanical testing.

### 2.1. Materials

#### 2.1.1. Materials Used for the RTM Process

The thermoplastic polyacrylate Elium^®^ 150 (Arkema, Colombes, France), with a density of 1.01 g/cc, a flexural modulus, flexural strength, and tensile strength of 3.250 MPa, 130 MPa, and 76 MPa, respectively, and a glass transition temperature of 130 °C was employed in the production of reinforced long glass fibre plates via the RTM process.

#### 2.1.2. Materials Used for Injection Overmoulding Process

Crastin^®^ LW9320 BK851 polybutylene terephthalate (Celanese Corporation, Irving, TX, USA), reinforced with 20% short glass fibres, with a density of 1.34 g/cm^3^, a flexural modulus and tensile strength of 7000 MPa and 120 MPa, respectively, a melting point of 220 °C and a glass transition temperature of 110 °C was used as the second component material employed for the overmoulding process.

### 2.2. Optimisation of Elium^®^ Composite Insert Manufacturing

#### 2.2.1. Resin Transfer-Moulded Elium^®^ Sheet Production

Elium^®^ sheets were prepared using an optimised laboratory resin transfer moulding process. This composite manufacturing technology is typically employed to produce high-quality, complex-shaped parts made of fabric-reinforced polymer matrices. The efficiency of the resin transfer moulding process lies in its ability to produce intricate and high-strength composite components with excellent surface finish, making it a preferred choice for various industries. In this instance, the process involved the use of a thermoplastic acrylic resin, Elium^®^, as the matrix and three-layer nonwoven fibreglass fabrics with acrylate-based coating as the reinforcement, with a density of 6.81 g/m^2^. The two-piece aluminium mould, with dimensions of 270 × 160 × 1 mm, was employed for the shaping of the final composite semi-products. The resin transfer moulding process comprises a number of key steps, each of which contributes to the overall success of the manufacturing process.

The initial stage of the RTM process involved the meticulous preparation of the mould, which was designed to ensure accurate reproduction of the part geometry that matched the desired dimensions of the final product, thereby achieving the intended specifications of the specimens used in subsequent injection moulding. Next, the fibreglass fabrics were fixed within the mould for integration with the thermoplastic acrylic resin. In contrast to some composite manufacturing processes, the removal of moisture from the reinforcement material was deemed unnecessary in this case. Consequently, the material was not subjected to a vacuum process for moisture elimination, thereby streamlining the overall manufacturing process. Finally, the reinforcing materials were dispensed into the mould using a single-component pressure vessel. This ensured controlled and uniform dispensing of the resin, facilitating the impregnation of the glass fibres and enhancing the overall quality of the final composite part. Once the mould was filled, the curing phase commenced. The applied curing conditions, namely a mould temperature of 80 °C and a curing time of 6 min, were defined based on preliminary optimisation experiments. During this curing period, the thermoplastic acrylic resin underwent a chemical reaction, transforming from a liquid to a solid state. This phase was critical for achieving the desired mechanical properties and structural integrity of the final composite part. Once the curing process was complete, the two-piece aluminium mould was opened, revealing the newly formed composite component. Samples of the desired shapes were cut from the prepared composite sheets by employing a non-contact approach using a laser beam [22]. In our case, samples with dimensions of 55 × 20 × 1 mm were cut from the prepared composite sheets at a controlled laser cutting speed of 15 mm/min^−1^ and 65% power using a 100 W BRM CO_2_ laser (BRM Lasers, Winterswijk, The Netherlands).

#### 2.2.2. Surface Pre-Treatment of Elium^®^ Inserts

The bonding area of the Elium^®^ inserts (peripheral parts with dimensions of 10 × 20 mm) was subjected to selected surface treatments before the overmoulding process to enhance the mechanical performance of the adhesion with the overmoulded material. Three different surface treatments were applied to the inserts. First, the Elium^®^ inserts were cleaned with isopropyl alcohol to remove any contaminants (grease) or impurities from the surface. This cleaning procedure serves as a reference for the subsequent comparison, which will be presented as an untreated specimen in the following section. The objective of the different surface treatments was to gain a deeper understanding of the impact of the surface treatment process on the mechanical properties of the final testing specimens. These treatments are referred to as solvent-induced swelling, atmospheric plasma jetting, and sandblast roughing. The specifics of these surface treatments are defined below:(a)Solvent-induced swelling—The bonding parts of the Elium^®^ inserts were immersed in toluene as an etching solution for 25 min. The treated inserts were then rinsed with distilled water and dried at room temperature for 10 min. During the 25-min immersion period, the toluene penetrates the surface of the Elium^®^ composite, causing controlled swelling. This swelling results in an increase in the thickness of the material. The solvent interacts with the polymer matrix, promoting molecular expansion and changing the structure of the surface. To assess the effectiveness of solvent-induced swelling, the thickness of the composites was measured both before and after the process. The measurements showed an increase in thickness between 0.11 mm and 0.125 mm, indicating the extent of swelling and modification achieved through the solvent treatment.(b)Atmospheric plasma treatment—In this process, the Elium^®^ composite inserts were treated with a plasma jet to activate and clean the bonding surface. The Plasma Beam PC (Diener Electronics GmbH, Ebhausen, Germany) was used at ambient temperature and atmospheric pressure. In this case, the use of two nozzles with a surface distance of approximately 12 mm was chosen as appropriate on the basis of preliminary tests. Figure 2 shows the schematic of the nozzle plasma system. It is possible to achieve completely clean and oxide-free surfaces by subjecting them to chemical attack with oxygen or air. The Elium^®^ composite surface was exposed to the plasma jet for periods of 5, 10, 15, and 30 s.

(c)Sandblasting—A surface modification process in which abrasive particles are propelled against the surface of a treated substrate to alter its surface properties and improve adhesion for subsequent processes, such as coating or bonding. Here, an SBC420 instrument (Reno-Tech s.r.o., Kaznejov, Czech Republic) was used to perform comprehensive abrasive treatment on the Elium^®^ substrate. The abrasive material used was slag (composition 30% SiO_2_, 40% AlO_3_;, and 30% CaO), with an average grain size of 120 µm (Sandblasting II) and an average grain size of 400–500 µm (Sandblasting I). The process was carried out at a pressure of approximately 0.2 MPa, with a substrate-to-nozzle distance of 8–10 cm at a perpendicular angle to the substrate surface for 30–40 s. The process is illustrated in Figure 3. The aim was to achieve variations in surface roughening, modification of mechanical interlock, and enhanced adhesion.

#### 2.2.3. Elium^®^ Insert Surface Characterization

(a)Scanning Electron Microscopy (SEM)—The surface of the substrate was observed using a Phenom XL G2 scanning electron microscope (Thermo Fisher Scientific, Waltham, MA, USA). Samples were analysed at an acceleration voltage of 10 kV in backscattered and secondary electron modes (50% mix).(b)Optical profilometry—Surface topographies were characterised using a 3D optical microscope, the Contour GT-K (Bruker Corporation, Billerica, MA, USA), based on white light interferometry with the use of 20× objective lenses. The resulting 2D and 3D topography maps were processed in the Gwyddion 2.55 software. Surface roughness values (Sa) and maximum height changes (Sz) were determined from five individual measurements.(c)Contact profilometry—The surface topography and roughness of all substrates were characterised using a DektaXT contact profilometer (Bruker Corporation, USA). A tip with a radius of curvature of 2 µm and a pressure equivalent to 3 mg was used. Surface roughness values (Ra) and maximum height changes (Rz) were determined from five individual measurements according to the SME B46.1 standard.

### 2.3. Optimisation of Overmoulded Composite Specimen Manufacturing

#### 2.3.1. Injection Insert Moulding of Two-Component Specimens

A Mitsubishi 180 Met III (Mitsubishi Heavy Industries, Ltd., Tokyo, Japan) electric moulding machine with a 46 mm diameter screw was employed to inject PBT melt in order to create the Elium^®^–PBT two-component specimens (see Figure 4). Prior to moulding, the Elium^®^ inserts were cleaned with isopropyl alcohol and allowed to dry for 50 s. The injection moulding conditions are listed in Table 1. The process was conducted with both treated and untreated Elium^®^ composites, incorporating three different surface modifications. Furthermore, different mould temperatures were employed based on the treatment conditions.

Untreated Elium^®^ composites were subjected to insert moulding at three different mould temperatures: 40 °C, 80 °C, and 120 °C. This temperature variation was employed to investigate the effect of mould temperature on the quality of the overmoulded parts and to identify the optimum processing conditions for untreated Elium^®^ composites. A higher mould temperature was not applied because the temperature of 140 °C used in preliminary tests was associated with the onset of technological problems related to the settling of the inserts in the mould. The selected melt temperature of up to 260 °C was employed to ensure sufficient melting of the Elium^®^ composite for effective overmoulding. During the injection phase, a pressure of 80 MPa was applied to force the melt material into the mould cavity, and a holding pressure of 50 MPa was maintained for 10 s to prevent any shrinkage or deformation. The parameters of injection temperature, pressure, and holding pressure parameters were kept constant throughout the overmoulding manufacturing process to ensure a consistent comparison between different mould temperatures.

In contrast, the treated samples were overmoulded at the constant mould temperature of 120 °C. This elevated temperature was chosen to achieve enhanced bonding between the Elium^®^ composite and the overmoulded PBT filled with 20% *w*/*w* of glass fibres. However, in this case, the effect of substrate surface modifications was evaluated.

The overall cycle time for the injection overmoulding process was 59.2 s, representing the total duration required for the mould to be filled, the material to cool and solidify, and the overmoulded part to be ejected.

#### 2.3.2. Mechanical Performance of Overmoulding Joining

The bonding performance was evaluated through the measurement of the shear lap strength between the Elium^®^ composite substrate and the PBT overmoulded part. This was carried out using an M350-5CT universal testing machine (Testometric Co., Ltd., Rochdale, UK) with a 10 kN load cell. A gauge length of 50 mm and a tensile rate of 2 mm/min were chosen for all measurements. For each moulding condition and tested surface modification, a total of six to eight specimens were evaluated for further comparison.

## 3. Results and Discussion

### 3.1. Surface Characterisation of Elium^®^ Inserts

Knowledge of the insert surface morphology allows us to better understand its influence on the bond strength between the inserts and the overmoulding material [24,25]. The modified Elium^®^ inserts used in this research were analysed using scanning electron microscopy, and the roughness profiles were recorded through profilometry analysis.

#### 3.1.1. Scanning Electron Microscopy

The surface microstructures of the Elium^®^ inserts, as recorded using SEM, are shown in Figure 5. While only trivial changes in the surface morphology were observed for the Elium^®^ inserts after solvent-induced treatment, the surface roughness of the sandblasted and plasma-treated inserts increased significantly. The grain size of the slag was smaller and, therefore, less aggressive for sandblasting II (average grain size of 120) compared to the larger average grain size used for sandblasting I (400–500 µm). In fact, sharp scratches and coarse erosion of the insert surface were observed for sandblasting I, in contrast to the milder eroded plates for sandblasting II. Furthermore, it can be observed that the plasma treatments resulted in a rather smooth removal of polymer material from the insert surface. Obviously, the longer the treatment, the more the polymer material is removed from the surface. This is due to the fact that increasing the plasma exposure time weakened the surface layer due to the higher energy involved, allowing the exposed glass fibres to emerge from the Elium^®^ surface. It should be emphasised that 30 s of plasma exposure not only resulted in fibre purging with an increase in the areas with clear stripped fibres (see Figure 5 and Figure 6) but also resulted in a reduction in the final bond strength. It should also be noted that the fibre surface after plasma treatment was clear and smooth, whereas after sandblasting, residual traces of the Elium^®^ matrix were observed, thus offering the potential for improved adhesion to the overmoulding material.

#### 3.1.2. Surface Roughness Analysis

In general, adhesion is strongly influenced by the morphological surface pore size and roughness. These variables can be altered by modifying the thermoplastic melt at the micron-scale roughness level of the substrate [26]. The presence of a rough topography increases the total area at the interface, facilitating polymer flow into the voids of the substrate (see Figure 1). While a graphical representation of the roughness profiles of treated substrates can be seen in Figure 6 and Figure 7, the relationship between surface roughness and defined shear strength can be checked in Figure 8. The exact surface roughness values obtained from profilometry experiments are given in Table 2.

Sandblasting I caused more roughness on the substrate than Sandblasting II and achieved more shear strength as well. Since plasma treatment typically results in a stronger bonding, it was selected as the preferred surface treatment technique [25], and it demonstrated the second-highest bond strength with 15 s of plasma treatment. However, Plasma 30 s had lower bond strength compared to Plasma 15 s, despite having higher roughness. The extended plasma exposure led to the uncoating of fibres because of the high energy of the plasma over a longer period. This uncoating of Elium^®^ from fibres on the substrate surface caused their release from the matrix, and consequently, this fact led to lower bonding strength and high deviation for various samples. As it is clear from Figure 6, while plasma treatment increased surface roughness through rather uniform polymer removal, causing the unfolding of the glass fibres from the Elium^®^ composite’s surface, the increased roughness caused by sandblasting was evidently incurred by casual hard particle blasting.

### 3.2. Mechanical Performance of PBT Overmoulded Elium^®^ Inserts

Successful bonding of the injection-moulded short glass fibre-reinforced PBT to the long glass fibre-reinforced Elium^®^ inserts was achieved under all processing conditions tested. Changing the conditions used did not result in any noticeable change in the appearance of the final bi-component samples under any of the conditions used. Following the lap shear tests, it was noted that no cohesive failure was observed in any of the samples tested, as there was no residue of PBT material on the Elium^®^ inserts.

#### 3.2.1. Evaluation of Mould Temperature Influence

A comparison of the mechanical performance of untreated samples at different mould temperatures (40 °C, 80 °C, and 120 °C) shows that higher mould temperatures generally result in increased bond shear strength (see Table 3). In fact, a doubling of the shear strength, increasing from 0.9 MPa to 1.8 MPa, was obtained when the mould temperature was increased from 40 °C to 120 °C. This can be attributed to improved material flow and intermolecular bonding due to the longer melt relaxation time at elevated temperatures.

The highest mould temperature of 120 °C, which ensures improved bond strength, was therefore used for overmoulding surface-treated inserts in order to evaluate the effect of the modifications applied.

#### 3.2.2. Evaluation of Surface Modification Effect

A comparison of shear strength between surface-modified Elium^®^ composite inserts and overmoulded glass-fibre-reinforced PBT parts manufactured at a constant mould temperature of 120 °C, as defined using tensile testing, is presented in Figure 8. Here, an average value of bonding strength with defined variations is introduced together with roughness (Ra) in order to clarify the impact of surface modifications. As can be seen, the application of suitable surface modifications has a significant effect on the bonding strength. As it is evident, while a slight increase in roughness associated with solvent-induced swelling (Ra was increased from value 0.47 to 0.6 μm) did not result in any changes in the bonding strength, a significant improvement was achieved through the sandblasting process, where the roughness was defined as 6 and 2.7 μm for Sandblasting I and II, respectively. It can be observed that the average bonding strength was tripled for sandblasted inserts, from a value of 1.8 MPa (Untreated inserts) to 5.4 MPa (Sandblasting I), with a standard deviation of 1.1 MPa, which is comparable to values obtained in studies focusing on defining the effects of overmoulding processing parameters [25]. Furthermore, it can be noted that surface modifications via solvent-induced swelling and sandblasting did not significantly alter the reproducibility of bonding strength, as the variations were not markedly elevated. Conversely, even plasma treatment applied for 15 s resulted in an increase in surface roughness to 12 μm, which was associated with an improvement in bonding strength to an average value of 4.9 MPa; this was accompanied by a significant reduction in the repeatability of the bonding strength. This increase in bond variation is likely related to the loss of fibre coating due to the applied plasma jet. With regard to the evaluation of the time required for plasma treatment, it is evident that there is an optimal time for its application. This may be connected to the sufficient time necessary to increase roughness, which is, on the other hand, limited by the release of glass fibres when the time of application is extended beyond a certain point. This was observed to be connected with the removal of fibres from the previous coating and polymer residues and accompanied by a decrease in bonding strength.

## 4. Conclusions

The aim of this research was to develop a hybrid thermoplastic bi-component product by combining a thermoplastic composite, namely Elium^®^ reinforced with fibreglass fabrics (70% *w*/*w*), with an engineering thermoplastic polymer, specifically 20% *w*/*w* short glass fibre-reinforced PBT, with optimised bond strength. The study and use of insert moulding were employed to overcome the adhesion challenge posed by the incompatibility of the materials used. The overmoulding process parameters and surface roughness produced through different surface treatments of Elium^®^ based composite inserts were investigated as potential solutions to improve the bonding strategy.

The results show that both the mould temperature during overmoulding and the surface modification of the inserts have a significant effect on the bond strength between the materials used. The first significant finding is that an increase in mould temperature during the overmoulding process has a positive correlation with bond strength; namely, a mould temperature of 120 °C doubles the bond strength to 1.8 MPa compared to 0.9 MPa at a mould temperature of 40 °C. The second important finding is that roughness alone is not responsible for good bonding, as improved bond strength of 5.4 MPa was achieved at 6 Ra-µm and 3.2 MPa at 27 Ra-µm. Although the surface roughness of the Elium^®^ inserts has a significant effect on the bond strength, probably due to the penetration of the thermoplastic melt into the substrate at the microscopic level, the activation of the insert surface by the application of an energy beam can be counterproductive by improperly removing the glass fibre surface coating.

The results presented in this paper highlight several points that merit further investigation in this area. In particular, the pore size and geometry of the pore structures should be explored and analysed in greater depth to gain a full understanding of the mechanism influencing interfacial bonding.

## Figures and Tables

**Figure 1 materials-17-03276-f001:**
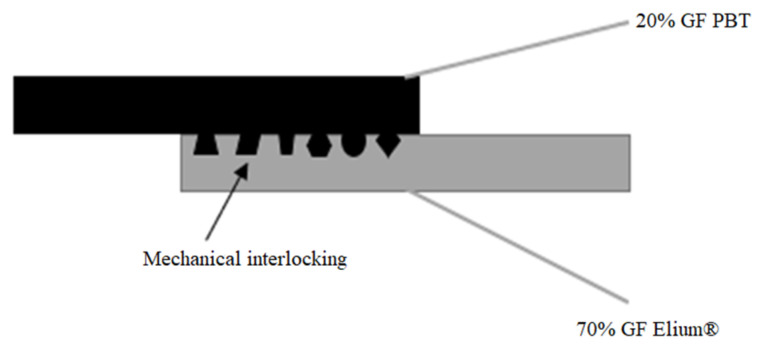
Schematic of mechanical interlocking via injection moulding direct joining [20].

**Figure 2 materials-17-03276-f002:**
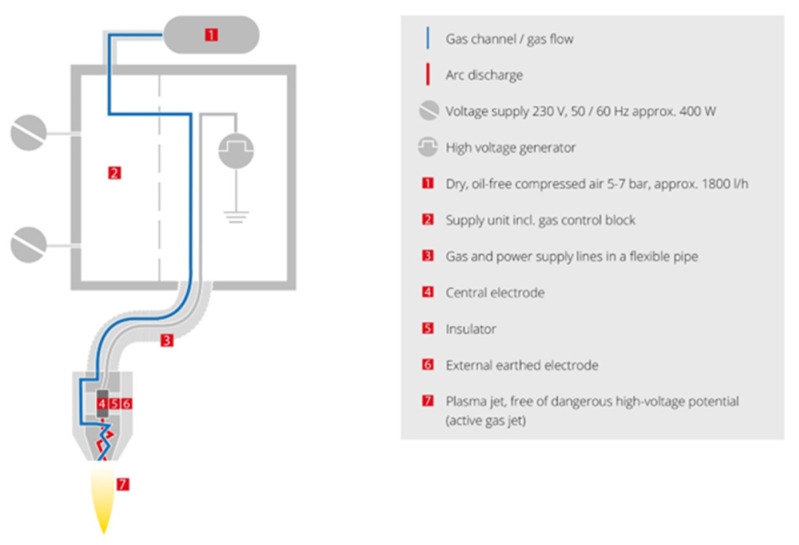
Scheme for the atmospheric pressure plasma system, as developed by Diner Electronics [23].

**Figure 3 materials-17-03276-f003:**
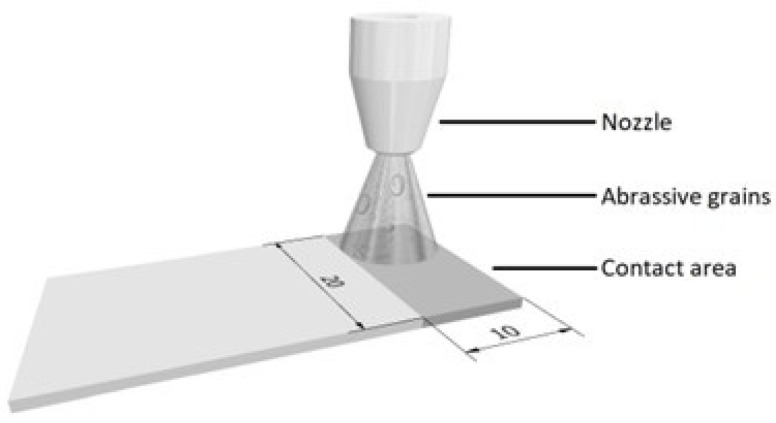
Schematic of sandblasting for Elium^®^ insert.

**Figure 4 materials-17-03276-f004:**
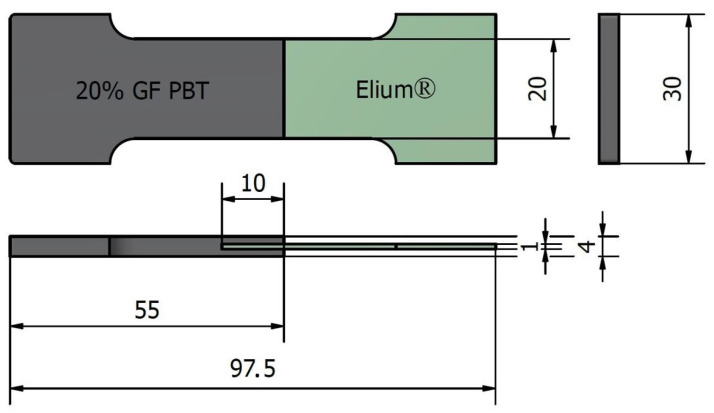
Bi-component specimen for the lap shear test.

**Figure 5 materials-17-03276-f005:**
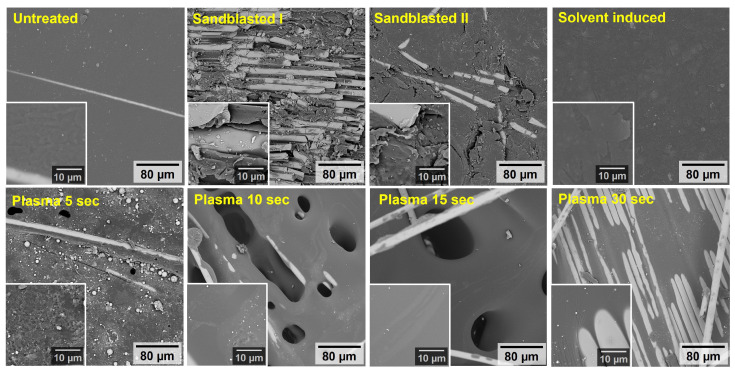
Scanning electron microscopy (SEM) images of the Elium insert surfaces for Untreated and other treated samples, namely Sandblasted I, Sandblasted II, Solvent-induced, Plasma 5 s, Plasma 10 s, Plasma 15 s, and Plasma 30 s.

**Figure 6 materials-17-03276-f006:**
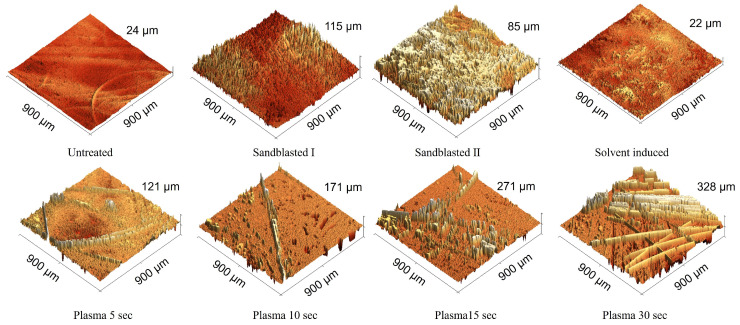
Three-dimensional (3D) images of the Elium insert surfaces for Untreated and other treated samples, namely Sandblasted I, Sandblasted II, Solvent-induced, Plasma 5 s, Plasma 10 s, Plasma 15 s, and Plasma 30 s, obtained with a 3D optical microscope.

**Figure 7 materials-17-03276-f007:**
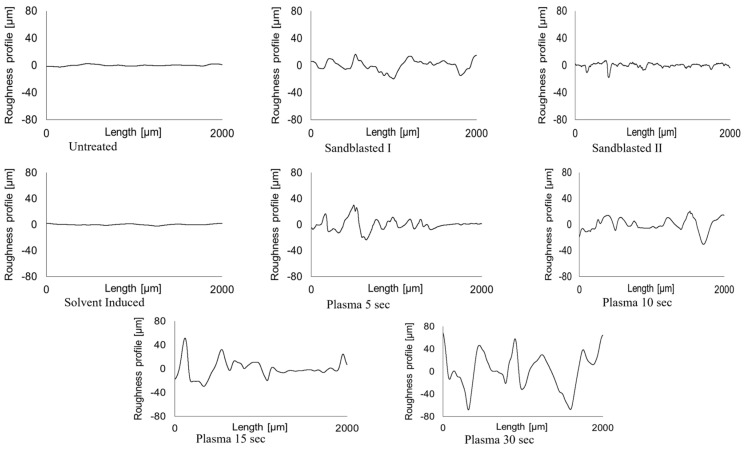
Surface roughness profiles of the Elium insert surfaces for Untreated and other treated samples, namely Sandblasted I, Sandblasted II, Solvent-induced, Plasma 5 s, Plasma 10 s, Plasma 15 s, and Plasma 30 s, obtained through contact profilometry.

**Figure 8 materials-17-03276-f008:**
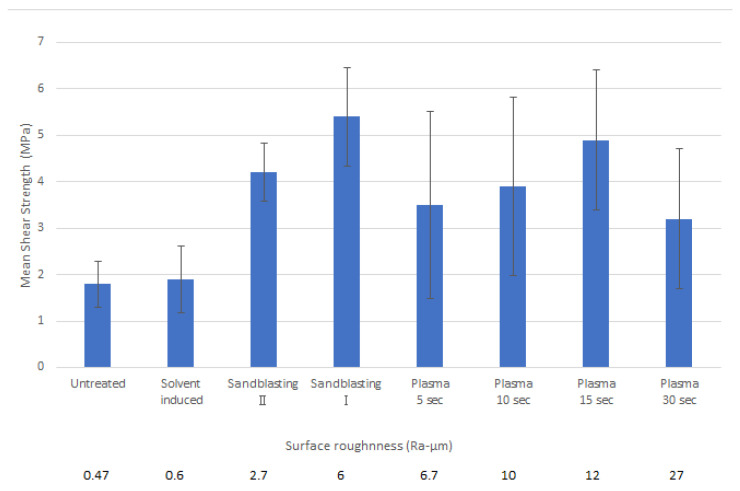
The mechanical performance of surface treatments on the bond strength between the Elium^®^ composite insert and the PBT overmoulded part of the bi-component samples, together with the results of surface characteristics defined using the average surface roughness (Ra).

**Table 1 materials-17-03276-t001:** Injection moulding processing parameters.

Injection speed	80 mm/s
Injection pressure	80 MPa
Injection unit temperature	220–260 °C
Nozzle temperature	260 °C
Holding pressure	50 MPa
Holding time	10 s
Cooling time	25 s
Mould temperature	40–120 °C

**Table 2 materials-17-03276-t002:** Average roughness values and their standard deviations caused by the utilised surface treatments, as defined via optical (Sa, Sz) and contact (Ra, Rz) profilometry.

Surface Treatment	Sa (µm)	Sz (µm)	Ra (µm)	Rz (µm)
Untreated	1.2 ± 0.2	18 ± 3	0.47 ± 0.05	2.9 ± 0.5
Sandblasting I	8 ± 0.5	100 ± 4	6 ± 0.3	27 ± 2
Sandblasting II	7.8 ± 0.4	123 ± 7	2.7 ± 0.03	18 ± 2
Solvent-induced	2.3 ± 0.4	65 ± 3	0.6 ± 0.03	2.6 ± 0.3
Plasma 5 s	7 ± 2	117 ± 12	6.7 ± 0.4	36 ± 2
Plasma 10 s	8.2 ± 0.7	172 ± 4	10 ± 2	50 ± 8
Plasma 15 s	10 ± 2	210 ± 17	12 ± 2	67 ± 9
Plasma 30 s	28 ± 2	303 ± 18	27 ± 4	131 ± 19

**Table 3 materials-17-03276-t003:** Shear strength of composite samples prepared from untreated Elium^®^ inserts and overmoulded with varying mould temperatures.

Mould Temperature (°C)	Mean Shear Strength (MPa)
40	0.9 ± 0.5
80	1.1 ± 0.5
120	1.8 ± 0.5

## Data Availability

Data contain within the article.

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
