# Peer review of "Surface Treatments’ Influence on the Interfacial Bonding between Glass Fibre Reinforced Elium® Composite and Polybutylene Terephthalate"

_materials, 2024, doi:10.3390/ma17133276_

Round 1

Reviewer 1 Report

Comments and Suggestions for Authors

In this work, a hybrid thermoplastic bicomponent was prepared by combining a thermoplastic composite, Elium® reinforced with glass fibres (70%), with a thermoplastic polymer specifically a 20% glass-filled PBT. The effect of surface treatments and moulding temperatures on the adhesion strength were investigated. The analysis of experimental results is somewhat simple, therefore, requires further in-depth discussion. Other comments are listed below:

 ï¼ˆ1)    Abstract, the first sentence, it is better to mention the kind of produces instead of “goods”.

(2)    The part of introduction is a bit long, however, seems incomplete. There should be a paragraph to point out the purpose and content of this work.

(3)    The highest shear strength was achieved at moulding temperature of 120 °C. Then what will happen if the temperature is further increased?

(4)    If we talked about shear strength, MPa should be used instead of N.

(5)    Table 4 and Figure 8 show the same result. Just keep one will be ok.

(6)    It is necessary to compare with results of reported similar materials.

(7)    It is better to show microstructure of the obtained samples.

(8)    General speaking, there should be no indicate figure in Conclusion. There are too many analysis sentences in Conclusion.

(9)    Please check the text. For example, Page 3-line 111, “hat” should be ”had”? Page 7-line 270, it is should be its.

Author Response

  1. Abstract, the first sentence, it is better to mention the kind of produces instead of “goods”.
  • We have made the suggested corrections as outlined in line 20.
  1. The part of introduction is a bit long, however, seems incomplete. There should be a paragraph to point out the purpose and content of this work.
  • With regard to the suggestion to add a paragraph to the introduction, we have taken this on board and have added a statement to the work’s purpose and content, which can be found on lines 119-125.
  1. The highest shear strength was achieved at moulding temperature of 120 °C. Then what will happen if the temperature is further increased?
  • The maximum mold temperature to heat up Elium insert described in the manuscripts was 120 ℃. In preliminary tests also higher temperatures were checked (namely 140°C), however if the temperature was increased to this level Elium insert started to make problems during their insertion and following overmoulding process. This finding is probably connected to the overcoming of glass transition temperature of Elium resin (aproximately 120°C). Text of manuscript was modified to mention this unwanted matter of fact. Please refer to lines 254-257.
  1. If we talked about shear strength, MPa should be used instead of N.
  • Changed to MPa in accordance with recommendations.
  1. Table 4 and Figure 8 show the same result. Just keep one will be ok.
  • Thank you for your comment. In order to keep the manuscript concise, Table 4 has been removed.
  1. It is necessary to compare with results of reported similar materials.
  • Thank you for your valuable input. We have incorporated comparison of our results with previous studies into the text, as you suggested. Please refer to lines 376-377
  1. It is better to show microstructure of the obtained samples.
  • In order to facilitate a more comprehensive understanding of the microstructure of surface-modified inserts, Figure 8 has been modified. The scale has been adjusted. Moreover the accompanying text has been revised to provide a more detailed comparison of SEM micrographs and profilometry. Please refer to lines 310-330 and Figures 5-7.
  1. General speaking, there should be no indicate figure in Conclusion. There are too many analysis sentences in Conclusion.
  • Following the recommendation to remove the indication of a figure in the conclusion, the conclusion has been modified with some quantitative results. Please refer to lines 403-416.
  1. Please check the text. For example, Page 3-line 111, “hat” should be ”had”? Page 7-line 270, it is should be its.
  • The entire manuscript was meticulously examined, and the authors are confident that all errors have been rectified.

Reviewer 2 Report

Comments and Suggestions for Authors

1.     In the Abstract section, please provide some important quantitative results and findings regarding the properties of thermoplastic composites. In addition, the mechanism of the effect of surface treatment on the properties of composites should be revealed.

2.     Introduction, it is recommended that a comprehensive comparison of thermosetting and thermoplastic composites should be conducted in terms of the production process and efficiency, mechanical properties, cost and some potential application scenarios. Furthermore, the authors should point out that the interface bonding between the thermoplastic resin and fiber needs be further improved due to the high viscosity of the resin and the poor wettability of the fiber. Some related research work can be reviewed to make corresponding supplements, such as Construction and Building Materials, 2024, 429: 136455. Materials, 2021, 14(17): 4861.  Polymers. 2022, 14, 1087.

3.     For the surface treatment, why surface treatment is conducted for the resin not the glass fiber? Please provide relevant explanation.

4.     The definition of some pictures in the paper is very poor, and it is recommended to replace them with high-definition pictures.

5.     It is recommended that the order of parts 2.4 and 2.5 be reversed. Authors should first give the preparation of the sample, followed by the relevant mechanical and microscopic properties of the test.

6.     What key information can be seen from the scanning electron microscope, Figure 5? It is recommended to add some keyword descriptions to the pictures.

7.     Table 3 shows that the shear strength with very large standard deviations, it is recommended to provide relevant explanations for this phenomenon. A similar situation applies to table 4.

8.     The improvement mechanism of shear strength by different treatment methods should be analyzed and discussed in detail. The current discussions are not enough to convince readers.

9.     It is recommended to include some important quantitative results in the conclusion section.

Author Response

  1. In the Abstract section, please provide some important quantitative results and findings regarding the properties of thermoplastic composites. In addition, the mechanism of the effect of surface treatment on the properties of composites should be revealed.
  • Thank you for your comment. We have added the quantitative results and revealed the bonding mechanism as recommended in line 30-33.
  1. Introduction, it is recommended that a comprehensive comparison of thermosetting and thermoplastic composites should be conducted in terms of the production process and efficiency, mechanical properties, cost and some potential application scenarios. Furthermore, the authors should point out that the interface bonding between the thermoplastic resin and fiber needs be further improved due to the high viscosity of the resin and the poor wettability of the fiber. Some related research work can be reviewed to make corresponding supplements, such as Construction and Building Materials, 2024, 429: 136455. Materials, 2021, 14(17): 4861.  2022, 14, 1087
  • It is a good point that “the interface bonding between the thermoplastic resin and fiber needs to be further improved due to the high viscosity of the resin and poor wettability of the fibre” But our research is more focused on obtaining the optimum bonding strength between Elium and PBT rather than the bonding of resin and fiber.
  1. For the surface treatment, why surface treatment is conducted for the resin not the glass fiber? Please provide relevant explanation.
  • Thank you for your comment. We are grateful to the glass fabric supplier for providing surface treatment of glass fabrics in order to achieve appropriate mechanical performance with acrylic Elium resin (this information was introduced in the manuscript in line 154). In our study, we were only focused on bonding between Elium composite substrate/inserts and overmoulding glass-reinforced PBT matrix. Thus, we believe that the problem with uncoated fibres was limited to the case when the longest time of plasma treatment was applied, which we felt was an unwelcome result. Please refer to lines 383-384.
  1. The definition of some pictures in the paper is very poor, and it is recommended to replace them with high-definition pictures.
  • We believe that, in light of your suggestion, the inappropriate figures legends (Figures 4-8) were modified.
  1. It is recommended that the order of parts 2.4 and 2.5 be reversed. Authors should first give the preparation of the sample, followed by the relevant mechanical and microscopic properties of the test.
  • In order to maintain the logical sequence and present the logical consequences of the research work, we would like to keep the order of the parts in Chapter 2 in the original organisation. However, the text introducing Chapters 2 was modified to express the progression of the work. Please refer to lines 129-133.
  1. What key information can be seen from the scanning electron microscope, Figure 5? It is recommended to add some keyword descriptions to the pictures.
  • The text of the manuscript describing the SEM images was reviewed and modified in order to present the results in a more concise manner. Please refer to lines 287-304.
  1. Table 3 shows that the shear strength with very large standard deviations, it is recommended to provide relevant explanations for this phenomenon. A similar situation applies to table 4.
  • Given that the results presented in Table 4 and Figure 8 were identical, we have taken the liberty of removing Table 4 to keep the manuscript as concise as possible.

Regarding large standard deviations in the results of mechanical performance of overmoulding of composite inserts by thermoplastics melt, it is important to note that a series of manufacturing free factors could potentially affect the outcome. These include the modification of inserts themselves, the insertion of composite substrate into the preheated mould, and the time and temperature of inserts preheating. These factors could be influenced by the length of mould opening, cooling/solidifying of thermoplastic matrix, and so on. To put it another way, it is to be expected that there will be a greater degree of variation in the results for the process that corresponds to the real industry process. 

  1. The improvement mechanism of shear strength by different treatment methods should be analyzed and discussed in detail. The current discussions are not enough to convince readers.
  • The manuscript was modified in accordance with this suggestion in order to clarify the achieved results and make them more clear. Please refer to lines - 342 to 390
  1. It is recommended to include some important quantitative results in the conclusion section.
  • In light of the valuable feedback, the conclusion chapter was modified accordingly. Please refer to lines 404-417.

Reviewer 3 Report

Comments and Suggestions for Authors

The manuscript by A. Matta et al. reports on the results of experiments attempting to improve interfacial bonding between two composite materials, namely fiber reinforced Elium and fiber reinforced polybutylene terephthalate. I came across the report and found it fairly trivial. In conclusion, the authors state that the objective of this "research was to develop a hybrid thermoplastic bi-component" by combining two composites. The only positive conclusion was made that increase of "a mould temperature" improved the bond strength about two times, which is i) trivial and ii) marginal improvement. Other findings are: "roughness was not solely responsible for good adhesion" and "several points worthy of further investigation in this area". There are no consideration about chemical composition of the surface upon plasma treatment and very rudimentary study of mechanical performance. I see no need to publish this report.

Comments on the Quality of English Language

English grammar is OK, but the style suffers from unnecessary repetitions, for example (lines 38-41) "Recent advancements in automotive applications have focused on thermoplastics (TP) and their composites, particularly organo-sheet composites[1]. Organo-sheet composites have attracted a lot of attention among them." Also (line 43): "Since the organic sheet matrix and the over-moulded TP are compatible, there are not many adhesion problems at the interface." Compatible it what sense? Line 66: what is Elium? Line 106: "The Elium® specimens prepared by RTM technology was over-moulded with PBT..." - should be "were". Line 111: "Secondly, the combination of Elium® and PBT hat the potential to offer enhanced..."

Author Response

The manuscript by A. Matta et al. reports on the results of experiments attempting to improve interfacial bonding between two composite materials, namely fiber reinforced Elium and fiber reinforced polybutylene terephthalate. I came across the report and found it fairly trivial. In conclusion, the authors state that the objective of this "research was to develop a hybrid thermoplastic bi-component" by combining two composites. The only positive conclusion was made that increase of "a mould temperature" improved the bond strength about two times, which is i) trivial and ii) marginal improvement. Other findings are: "roughness was not solely responsible for good adhesion" and "several points worthy of further investigation in this area". There are no consideration about chemical composition of the surface upon plasma treatment and very rudimentary study of mechanical performance. I see no need to publish this report.

  • The reviewer's comments have been carefully considered and the manuscript has been improved by clarifying the logical organisation of the flow text. We have tried to present the processing parameters of overmoulding (mould temperature) and insert modifications (surface treatments) more separately. Hopefully, the modified descriptions of the results section and conclusions will now give a clearer idea of the results obtained. With regard to the concern about the precise definition of the chemical composition of the inserts after plasma treatment, the authors respectfully propose that, given the applications of atmospheric plasma jets (a common industrial tool for in-line surface activation) have been defined as counterproductive for bonding variations of materials used in this manuscript, the use of a suitable tool for its evaluation (for example XPS) may also be counterproductive and may not necessarily lead to improvements in the manuscript results.

English grammar is OK, but the style suffers from unnecessary repetitions, for example (lines 38-41) "Recent advancements in automotive applications have focused on thermoplastics (TP) and their composites, particularly organo-sheet composites[1]. Organo-sheet composites have attracted a lot of attention among them." Also (line 43): "Since the organic sheet matrix and the over-moulded TP are compatible, there are not many adhesion problems at the interface." Compatible it what sense? Line 66: what is Elium? Line 106: "The Elium® specimens prepared by RTM technology was over-moulded with PBT..." - should be "were". Line 111: "Secondly, the combination of Elium® and PBT hat the potential to offer enhanced..."

  • The entire manuscript was meticulously examined, and the authors are confident that all errors have been rectified.

Round 2

Reviewer 1 Report

Comments and Suggestions for Authors

The revised manuscript can be accepted now.

Author Response

We would like to express ou gratitude for your kind assistance. 

Reviewer 2 Report

Comments and Suggestions for Authors

The authors have provided some modifications; however, the following comments should be addressed further.

Question 1: Some quantitative results on bond strength have not been added in the revised version.

Question 2: The comments: “A comprehensive comparison of thermosetting and thermoplastic composites should be conducted in terms of the production process and efficiency, mechanical properties, cost and some potential application scenarios.” has not been addressed.

Question 4: Although some efforts have been made, the clarity of the pictures is not enough, such as figure 4.

Question 5: Please provide the detailed explanation on the order of parts 2.4 and 2.5.

Question 7: There are still very large standard deviations in figure 8. The reasonable explanation is necessary.

Author Response

Question 1: Some quantitative results on bond strength have not been added in the revised version.

With regard to the information about the quantitative increase of bond strength, we have taken the liberty of implementing it into the abstract (please see lines 33-34).

Question 2: The comments: “A comprehensive comparison of thermosetting and thermoplastic composites should be conducted in terms of the production process and efficiency, mechanical properties, cost and some potential application scenarios.” has not been addressed.

The authors have taken the liberty of modifying and extending the introduction in order to provide a more comprehensive comparison of thermosetting and thermoplastic composites. They have specified production technologies in greater detail, together with the mechanical incidents and economic aspects connected to the used technologies. However, it is necessary to stress that only general conclusions can be presented here, since specific products and companies' technological potential significantly affect the final practical impact according to specific applications scenarios, which are presented here as well.

Question 4: Although some efforts have been made, the clarity of the pictures is not enough, such as figure 4.

Accept our apologies for the oversight. The quality of Figure 4 has been improved.

Question 5: Please provide the detailed explanation on the order of parts 2.4 and 2.5.

We would like to propose some changes to the order of the manufacturing steps, including a new diversification of Chapter 2. We believe it is important to describe the evaluation of each processing technology immediately, since without this it is not possible to proceed with the subsequent manufacturing step. We have divided Chapter 2 into three sections: a materials description part (2.1.), a section on Elium insert manufacturing (2.2.) and a section on overmoulded specimens manufacturing (2.3.).

Question 7: There are still very large standard deviations in figure 8. The reasonable explanation is necessary.

It seems that the standard deviations defined in our experiments, 1.1 MPa for the sandblasting surface treatment, are consistent with the manufacturing method used for the final bi-component specimens, namely the overmoulding technique. This information, based on a comparison with several previous research works carried out (references 26-28), is stated also in the manuscript (see line 410). It would be beneficial to consider this when applying this method in industry practice.

Reviewer 3 Report

Comments and Suggestions for Authors

The second edition of the manuscript submitted by A. Matta et al. has been improved in regard to presentation quality including clearer thoughts expression, better quality images, and scientific soundness. The design and number of experiments is still the same lacking depth of comprehension and analysis of facts including chemical composition of matrix of Elium. I believe that the overall merit of the manuscript is still rather low. Keeping this in mind, I may not recommend publication of the manuscript.

Author Response

Review report 3

The second edition of the manuscript submitted by A. Matta et al. has been improved in regard to presentation quality including clearer thoughts expression, better quality images, and scientific soundness. The design and number of experiments is still the same lacking depth of comprehension and analysis of facts including chemical composition of matrix of Elium. I believe that the overall merit of the manuscript is still rather low. Keeping this in mind, I may not recommend publication of the manuscript.

We would like to thank the reviewer for kind words about the quality improvement of our work after review reports were considered. As our focus was on defining effects that could be easily changed in the technological process – specifically, changing surface roughness and processing parameters – we would like to apologise for not having evaluated the effect of changing the chemical compositions of the commercially accessible materials used in our study, namely polyacrylate Elium and polybutylene terephtalate Crastin, in more detail. Nevertheless, we believe that the results presented will be valuable for industry practice.